# Research Progress of Flavonoids Regulating Endothelial Function

**DOI:** 10.3390/ph16091201

**Published:** 2023-08-23

**Authors:** Hao Li, Qi Zhang

**Affiliations:** The Basic Medical College, Shaanxi University of Chinese Medicine, Xianyang 712046, China; 221010011697@sntcm.edu.cn

**Keywords:** flavonoids, endothelial dysfunction, oxidative stress, NO, bioavailability

## Abstract

The endothelium, as the guardian of vascular homeostasis, is closely related to the occurrence and development of cardiovascular diseases (CVDs). As an early marker of the development of a series of vascular diseases, endothelial dysfunction is often accompanied by oxidative stress and inflammatory response. Natural flavonoids in fruits, vegetables, and Chinese herbal medicines have been shown to induce and regulate endothelial cells and exert anti-inflammatory, anti-oxidative stress, and anti-aging effects in a large number of in vitro models and in vivo experiments so as to achieve the prevention and improvement of cardiovascular disease. Focusing on endothelial mediation, this paper introduces the signaling pathways involved in the improvement of endothelial dysfunction by common dietary and flavonoids in traditional Chinese medicine and describes them based on their metabolism in the human body and their relationship with the intestinal flora. The aim of this paper is to demonstrate the broad pharmacological activity and target development potential of flavonoids as food supplements and drug components in regulating endothelial function and thus in the prevention and treatment of cardiovascular diseases. This paper also introduces the application of some new nanoparticle carriers in order to improve their bioavailability in the human body and play a broader role in vascular protection.

## 1. Introduction

Cardiovascular disease (CVD), which includes coronary artery disease, stroke, hypertension, heart failure, rheumatic etiology/congenital heart disease, and peripheral vascular disease, is the leading cause of death worldwide, causing about 17.3 million deaths per year and showing a sustained growth trend [1]. Endothelial dysfunction in the vascular system interacts with the pathogenesis of CVDs and often becomes the initial stage of vascular disease [2].The disruption of the endothelium-dependent vasodilator–contractor balance directly or indirectly affects most CVDs, such as hypertension, coronary artery disease, chronic heart failure, peripheral artery disease, and diabetes mellitus [3].The arterial wall is composed of three layers from the outside to the inside, namely the adventitia, media, and intima. Vascular endothelial cells (VECs) are a layer of flat squamous cells that continuously cover the surface of the vascular lumen and form the intima layer together with connective tissue [4] (Figure 1). The endothelium is often in a metabolically active state. As the hub of the cell network, it is not only an important barrier between the blood and the vessel wall but also secretes a variety of vasoactive factors according to the local environment to affect the balance between vasodilation and contraction responses. Endothelium-dependent vasodilators mainly involve NO, endothelium-derived hyperpolarizing factor (EDFF), and prostacyclin (PGI2), while endothelium-dependent vasoconstrictor responses are mainly associated with endothelin-1 (ET-1), angiotensin II (Ang II), reactive oxygen species (ROS), and thromboxane A2 (TXA2) [5]. Of these, NO is a key active substance to maintain vascular homeostasis. The decrease of NO bioavailability, including the decrease of NO production and/or the increase of NO degradation by superoxide anion, marks the beginning of endothelial dysfunction. Oxidative stress induced by CVD-related events often inhibits the bioavailability of NO and induces chronic vascular inflammation, which aggravates endothelial damage.

For a long time, people have gradually realized that diet plays an important role in the etiology of many chronic diseases, causing differences in the incidence and mortality of chronic diseases among populations in different countries and regions of the world. Therefore, it is crucial to prevent CVD by adjusting the diet. Based on this, the traditional Mediterranean diet (MedDiet), rich in vegetables, fruits, nuts, beans, etc., has been identified as one of the healthiest CVD-preventing diets [6]. One of its main active ingredients, polyphenols, is a secondary metabolite of plants and a major source of antioxidants in the diet. According to the molecular skeleton structure, it can be divided into flavonoids and non-flavonoids such as phenolic acids, stilbenes, phenolic alcohols, lignans, etc. [7]. Thus far, more than 8000 polyphenols have been known, of which more than 5000 are flavonoids [8]. Within each category, there is considerable heterogeneity in the number and position of substituents such as hydroxyl(OH), methoxy(OCH3) and sugar groups, which determine the physicochemical properties and biological activity of polyphenols. For example, the position and degree of hydroxylation significantly affect their antioxidant properties [9]. Numerous epidemiological studies have shown that moderate intake of dietary polyphenolic compounds may contribute to the prevention of atherosclerosis (AS), arterial hypertension, and coronary heart disease (CHD) [9]. Previous studies have shown that intake of foods with high dietary flavonoid content is negatively correlated with cardiovascular disease mortality, and its pharmacological activity may be related to food type, intake dose, and in vivo bioavailability [10]. In addition, as natural products widely found in nature, flavonoids are also extremely abundant in Chinese herbs and have been proven to achieve the function of preventing and controlling CVD through multi-targeting and multi-pathway effects [11,12].

Currently, traditional antihypertensive drugs and nitric oxide substitutes are commonly used to improve cardiovascular diseases. Considering the inevitable side effects and drug tolerance, new anti-inflammatory agents and antioxidants based on phytochemicals have attracted increasing scientific interest [13]. Among many natural active molecules, flavonoids have received special attention due to their wide range of biological activities. However, the detailed mechanism of protecting endothelial cells (ECs) is still unknown. This review focuses on the literature in recent years and preliminarily summarizes the performance and potential mechanism of flavonoids and polyphenols in improving endothelial dysfunction and cardiovascular health and discusses their bioavailability after metabolism in the human body.

### 1.1. Endothelial Function and Vascular Homeostasis

Known as the gatekeeper of vascular biology due to their location at the critical interface between circulating blood and the cellular environment, the ECs’ surface is covered with a polysaccharide calyx, formed by negatively charged glycoproteins, proteoglycans, and glycosaminoglycans, which can inhibit platelet and leukocyte adhesion and promote vascular barrier permeability. It can also mediate shear-stress-induced NO release and exert vascular protection [14]. NO is produced by the oxidation of L-arginine by endothelial nitric oxide synthase (eNOS) with the help of the cofactor tetrahydrobiopterin (BH4) [15]. It then diffuses into subcutaneous vascular smooth muscle cells and triggers cyclic guanosine monophosphate (cGMP)-dependent vasodilation through activation of guanylate cyclase (SGC) [16]. Cyclooxygenase (COX)-derived PGI2 stimulates the prostacyclin receptor and activates adenylate cyclase (AC) in smooth muscle cells and then activates the cyclic adenosine monophosphate/protein kinase A (cAMP/PKA) signaling pathway to reduce Ca2+-mediated vascular tone (Figure 2). Several vasoconstrictor molecules produced by the endothelium, such as ET-1, Ang-II, and TxA2, tend to be released to regulate platelet activity, coagulation cascade reactions, and the fibrinolytic system under physiological conditions. Under physiological conditions, ECs constitute a non-adhesive surface that prevents platelet activation and coagulation cascade reactions. After vascular injury, endothelial activation is involved in all subsequent major hemostatic events and restricts clot formation to specific areas where hemostasis is required as well as restoration of vascular integrity [17]. There are also mechanosensors/mechanosensitive complexes on the surface of ECs that sense shear stresses generated by blood flow and convert them into biochemical signals that regulate vascular tone and homeostasis in vivo and intervene in NO production to achieve vascular homeostasis through a variety of mechanisms [18]. ECs also have metabolic activity, maintaining their proliferation and vasodilation functions through amino acid (AA) metabolism. Endothelial glutaminase can catalyze the metabolism of glutamine to α-ketoglutarate, induce endothelial senescence, and inhibit endothelial cell proliferation through pharmacological effects [19]. The functions of healthy endothelium, such as dynamic maintenance of vascular tension, anti-oxidation, anti-thrombosis, anti-inflammation, and participation in vascular metabolism, are of great significance to the homeostasis of cardiovascular system and have great reference value for the preclinical testing of new drugs (Figure 3).

### 1.2. Endothelial Dysfunction

#### 1.2.1. Oxidative Stress and eNOS Uncoupling

A healthy endothelium enables the regulation and maintenance of vascular homeostasis, and any disturbance to this delicate and valuable balance can lead to the development of endothelial dysfunction. As mentioned earlier, NO is an endothelium-derived relaxing factor that, in addition to regulating vasodilatation, affects vascular homeostasis through a variety of pathways, such as inhibition of smooth muscle cell proliferation, platelet aggregation, adhesion of platelets and monocytes to endothelial cells, LDL oxidation, expression of adhesion molecules, and endothelin production. However, increased oxidative stress or reduced antioxidant enzyme activity, for example, can lead to reduced NO bioavailability [20,21].

Oxidative stress may be caused by the excessive production or accumulation of free-radical-reactive substances such as reactive oxygen species (ROS), reactive nitrogen species (RNS), and reactive sulfur species (RSS) [22]. Among them, ROS includes molecules such as H_2_O_2_, superoxide anion (O_2_^•−^) and hydroxyl radical (•OH), which are key molecules in maintaining the redox state of cells and physiological signaling [23]. Oxidative stress is usually defined as a pathological state caused by the imbalance between prooxidants and antioxidants. The pro-oxidants that have been found include NADPH oxidase (NOX), xanthine oxidase (XO), mitochondrial respiratory chain enzymes, and dysfunctional eNOS. The antioxidant enzyme system includes superoxide dismutase (SOD), catalase (CAT), glutathione peroxidase (GPx), heme oxygenase (HO), thioredoxin reductase (Trx), and paraoxonase (PONs) [15]. There is evidence that oxidative stress induced by a large amount of ROS in the vascular wall will aggravate endothelial dysfunction, and CVD-related events such as diabetes, hypertension, dyslipidemia, smoking, or obesity are often important inducing factors [24].

Under oxidative stress conditions, eNOS removes an electron from NADPH and donates it to O_2_ to generate O_2_^•−^ rather than NO, and this process is known as eNOS uncoupling [25]. BH4 has an important auxiliary role for efficient electron transfer in the eNOS-catalyzed cycle. In the process of oxidative stress, O_2_^•−^ reacts with NO to produce peroxynitrite (ONOO-), which can rapidly oxidize BH4 to BH2. The decrease in the BH4/BH2 ratio leads to the fact that electrons cannot be transferred to the N-terminal oxygenase domain of other eNOS monomers, thus exacerbating eNOS uncoupling [26]. Studies have shown that NOX-derived H_2_O_2_ in endothelial cells down-regulates the expression of DHFR in response to angiotensin II, which also leads to the lack of BH4 and the decoupling of eNOS [27]. In addition, oxidative stress has been shown to promote the synthesis of asymmetric dimethyl-l-arginine (ADMA), which competes with the eNOS substrate l-arginine [28], thereby inhibiting eNOS synthesis of NO. Oxidative stress can also promote the S-glutathionylation of eNOS to reversibly reduce the production of NO and increase the production of O^2−^ [29].

The substrates and cofactors in the process of NO synthesis by eNOS are affected by oxidative stress, and the products of eNOS uncoupling promote the production of ROS, resulting in a vicious circle that drastically reduces NO bioavailability, severely disrupts the endothelial environmental homeostasis, and leads to a cascade of cardiovascular diseases.

#### 1.2.2. Inflammation

Inflammation has been reported to play an important role in all stages of the AS process, with vascular inflammation being the process that leads to alterations in the vascular wall and subsequently to endothelial dysfunction [30]. Chronic inflammation may be caused by a variety of stimuli, such as oxidative stress, pro-inflammatory cytokines, hypercholesterolemia, hypertension, and shear stress [31]. When damaged, ECs are activated and produce inflammatory factors such as interleukin-8 (IL-8), chemokines, colony-stimulating factors, interferons, monocyte chemotactic protein-1 (MCP-1), intercellular adhesion molecule-1 (ICAM-1), p-selectin, e-selectin, and vascular adhesion molecule-1 (VCAM-1). These substances attract monocytes and neutrophils, which attach to activated ECs and penetrate the arterial wall, thereby triggering inflammation [32].

O^2−^ and other ROS generated by oxidative stress stimulate nuclear factor kappa B (NF-κB), which in turn activates various pro-inflammatory cytokines such as tumor necrosis factor-α (TNF-α) and IL-1. TNF-a and IL-1β can stimulate endothelial cells to secrete other pro-inflammatory cytokines (IL-6), which in turn stimulate hepatocytes to produce and release a variety of acute phase reactants, including fibrinogen and C-reactive protein, regulating chronic inflammation and acute-phase response [33]. The circulating inflammatory markers C-reactive protein and IL-6 are able to up-regulate the production of tissue factor (TF) and vascular hemophilic factor (vWF), while inhibition of the expression of thrombomodulin, NO, and PGI2 allows for a shift in the endothelial milieu from an antithrombotic to a prothrombotic state [17]. Endothelial dysfunction can also be induced directly via oxidative low-density lipoprotein (LDL) receptor-1 [34]. At the same time, TNF-α in turn activates NF-κB to increase the expression of cell adhesion molecules [35]. In addition, TNF-α was found to up-regulate NOX activity in endothelial cells and increase O^2−^ levels in the vessel wall [36]. Such a positive feedback accelerates monocyte adhesion to the endothelium, leading to chronic inflammation of the vessel wall. Other factors such as oxidized low-density lipoprotein (ox-LDL) can stimulate the secretion of adhesion molecules by ECs and trigger the formation of vascular lesions. Ang II, a key effector of RAS, activates NF-κB, which up-regulates the expression of inflammatory cytokines and adhesion molecules exerting inflammatory effects and promoting the development of AS plaques [37]. It also activates their vascular G-protein-coupled receptors, leading to NOX activation and increased ROS production [38]. Lipopolysaccharide (LPS) can activate immune cells (such as macrophages) to secrete inflammation-mediated cytokines to catalyze inflammation. It can also activate ERK1/2, JNK, and p38 MAP kinases (MAPKs), which ultimately regulate the activity of transcription factor NF-kB and regulate the expression of inflammatory mediators such as inducible nitric oxide synthase (iNOS), cyclooxygenase-2 (COX-2), TNF-α, IL-1α, IL-1β, and IL-6 [39,40]. Vascular inflammation is one of the major disruptors of the vascular homeostatic environment and endothelial normal physiology, and the NF-кB and MAPK pathways are closely related to the secretion of pro-inflammatory cytokines, which deserves to be further explored in the exploration of anti-inflammatory therapeutic strategies in the future.

In addition to the typical endothelial dysfunction-causing factors described above, disruption of vascular homeostatic balance by other pathways is of equal interest, for example, the endothelial-to-mesenchymal transition (EndoMT) by TGF-β signaling, in which ECs lose endothelial features but acquire mesenchymal-like morphology and gene expression patterns [41]. In addition, in the development of AS, endothelial-dependent permeability and vasodilation are often inhibited by VECs death, especially the regulatory death of endothelial cells, such as ferroptosis and autophagy [42]. As one of the initial inducing factors of AS, ferroptosis promotes the collapse of cell membrane and mitochondrial membrane, resulting in endothelial injury and death. It can also induce intravascular plaque formation, and the resulting vascular remodeling is a key factor in the stability of plaque in advanced AS [43]. Excessive autophagy caused by severe oxidative stress or inflammatory stimulation causes autophagy-dependent cell death and destroys plaque stability [44]. Autophagy of endothelial cells causes VCAM-1, ICAM-1, and other levels to shift down, while the infiltration of macrophages and foam cells increases, which also promotes arterial thrombosis [45] (Figure 4).

In fact, there are often synergistic or promoting effects between many factors that cause endothelial dysfunction, which have a superposition effect on disrupting vascular homeostasis and accelerating the occurrence and development of CVDs. Through the in-depth study of the mechanism of endothelial function/disorder, we can accelerate the discovery of new therapeutic drugs and implement effective targeted therapy.

## 2. Flavonoids

### 2.1. Sources, Classification, and Chemical Properties of Flavonoids

Flavonoids are the most abundant and widely studied natural phenolic compounds, which are commonly found in fruits, vegetables, wine, tea, and Chinese herbal medicine [46]. Flavonoids have a basic C6-C3-C6 15-carbon skeleton consisting of two aromatic rings and a pyran ring. They are classified into six subclasses based on their carbon structure and oxidation level, namely flavones, flavonols, flavanones, flavan-3-ols (flavanols), isoflavones, and anthocyanins [47] (Figure 5 and Figure 6). Dietary flavonoids in nature exist in the form of glycosides such as glucosides, galactosides, arabinosides, rhamnosides, and rutinosides [48]. All dietary flavonoids except flavanols exist in glycosylated forms [49], and deglycosylation is a key step in the absorption and metabolism of flavonoid glycosides [50]. In nature, flavonoids usually do not exist alone. Oral intake of dietary flavonoids can also interact with other compounds, such as carbohydrates, fats, proteins, acids, etc., and their physiological activity may also change [51]. When flavonoids derived from Chinese herbal medicines are administered orally, their effectiveness as therapeutic drugs is seriously reduced due to their poor solubility, low permeability, and poor stability. For example, the 2,3-position double bonds of flavonoids and flavonols easily form a planar structure, resulting in tight molecular arrangement, so it is difficult for solvent molecules to penetrate their molecular structure [52]. Despite this, flavonoids have been shown to exert a wide range of pharmacological activities and involve multiple signaling pathways to achieve antioxidant, anti-inflammatory, anti-hyperlipidemic, and cardioprotective effects [53] (Table 1).

### 2.2. Bioavailability of Flavonoids

As mentioned earlier, although epidemiological studies have demonstrated the ability of long-term, high intake of flavonoid-rich foods to reduce the incidence of CVD events, their bioavailability in the human body is actually very low due to differences in composition, subclasses, glycosylation, molecular weight, and esterification [87]. After oral ingestion, dietary flavonoids are first metabolized through the detoxification pathways of exogenous substances and drugs; however, the ability of saliva and gastric acid to modify the structure of the compounds in a primary way is limited and minor [50]. Due to the relatively small area of the gastric mucosa, it also limits its absorption capacity. Several types of flavonoids known to be absorbed by the human body through the stomach are quercetin, genistein, and daidzein [88]. The small intestine is a key part of drug absorption and metabolism. Flavonoid aglycones are hydrophobic and have a small molecular structure. They can be directly absorbed by villous epithelial cells on the small intestine wall through passive diffusion [89]. The absorption rate of flavonoids is different due to the influence of structure and pH. Studies have shown that acidic media are more conducive to the passage of flavonoids through the Caco-2 cell model [90]. There are two main pathways known for the absorption of flavonoid glycosides in the small intestine. One is that flavonoid glycosides can be hydrolyzed to glycosides by lactase-phlorizin hydrolase (LPH) at the edge of the mammalian small intestine [91]. The other pathway is that flavonoid glycosides hydrolyzed to glycosides by broad-spectrum β-glycosidases (e.g., cytosolic β-glucosidase (CBG)) are able to exert deglycosylation to convert sugar-bound polyphenols to glycosidic ketones, which are then transported via the sodium-glucose co-transporter type 1 (SGLT1) to intestinal epithelial cells, where the phase II enzyme produces the corresponding affixed metabolites, which finally enter the circulatory system as glycosides or couplers. [92,93]. Three types of phase II enzymes that have been reported to be present in intestinal epithelial cells are uridine-5′-diphosphate-glucuronosyltransferase (UGT), sulfotransferase (SULT), and catechol-O-methyltransferase (COMT). In humans, both UGT and SULT are thought to contribute to monoglucuronide and sulfate production [94,95].

The liver is another important site for the phase II metabolism of flavonoids, where there are two main forms of metabolism: oxidation and conjugation reactions. Oxidation reaction mainly relies on cytochrome P450 enzymes in the liver to metabolize flavonoids [96]. The binding reaction mainly refers to the action of various catalysis enzymes, prompting the phase I metabolites containing some polar functional groups (such as hydroxyl) and some endogenous substances coupled or combined to produce a variety of binding products [97]. The conjugates are further metabolized by sulphation and methylation, after which they either enter the blood circulation or return to the digestive tract via the hepatic–intestinal circulation [98]. Although this enterohepatic circulation contributes to higher plasma levels and half-lives of flavonoids, the total intake of dietary polyphenols in the small intestine is only approximately 10% [99], with the remaining metabolites being transported to the large intestine.

The gut microbiota is able to deconstruct the original compounds into more readily absorbed flavonoid metabolites through several intertwined steps of ester and glycoside hydrolysis, demethylation, dehydroxylation, and decarboxylation. Metabolites produced in the large intestine are reabsorbed for further phase II metabolism at the local and/or liver levels and eventually excreted in large quantities into the urine [100]. Indeed, polyphenols, including flavonoids, are able to exert a prebiotic effect on the intestinal flora, and proanthocyanidins in particular have an inhibitory effect on the progression of lifestyle-related diseases such as diabetes by altering the pattern of the intestinal microbiota [101] (Figure 7).

The ability of dietary flavonoids to actually exert vasoprotective effects after in vivo metabolism is limited, and further exploration of the dependency relationship with gut flora may be a new idea to improve their bioavailability.

## 3. Protective Effects of Flavonoids on Endothelial Cells

### 3.1. Flavones (Luteolin and Baicalin)

Flavones are widely distributed in the plant kingdom, including in parsley, celery, red peppers, and various herbs. In contrast to other subunits, they have a double bond between C2 and C3 in the flavonoid backbone, are not substituted at the C3 position, and are oxidized at the C4 position [102]. We next discuss two well-studied flavones: luteolin and baicalin.

Luteolins are widely found in a variety of food and medicinal plants, including artichoke (*Cynara cardunculus*), which is one of the world’s best-known “medicinal” plants, with a long history of medicinal therapeutic use [103]. In terms of vascular relaxation, studies have shown that luteolin can directly act on vascular ECs in a dose-dependent manner (10–100 μmol/L), leading to rapid activation of eNOS and production of NO, resulting in relaxation of vascular tension in rat aortic rings and playing the same mechanism in primary human aortic endothelial cells (HAEC) [54]. By down-regulating the HIF-2α-Arg-NO axis and promoting the PI3K-AKT-eNOS-NO signaling pathway, luteolin regulates the NO content in the lungs of hypoxic pulmonary hypertension (HPH) rats and the supernatant of pulmonary artery endothelial cells (PAECs), thereby improving hypoxia-induced pulmonary hypertension [54]. In H_2_O_2_-induced injury of human umbilical vein endothelial cells (HUVECs), luteolin (2.5–20 mM) was able to regulate the AMPK/PKC pathway [55] and inhibit the ROS-mediated activation of the P38 MAPK/NF-kB signaling pathway [56], respectively, thereby exerting anti-oxidative stress properties. Literature studies in recent years have shown that luteolins also possess significant anti-inflammatory properties. In vitro experiments have shown that physiologically achievable concentrations (0.5μM–2 μM) of luteolin can effectively inhibit TNF-α-induced expression of MCP-1, ICAM-1, and VCAM-1; block monocyte adhesion to ECs; and ameliorate vascular endothelial inflammation through inhibition of the IΚBα/NF-κB signaling pathway [57]. For microvascular ECs, it increases cAMP levels and inhibits ERK phosphorylation, thereby inhibiting lymphocyte function-associated antigen-1 (LFA-1) expression in neutrophils and exerting an anti-inflammatory effect by attenuating adhesion to the endothelium [104]. Luteolin-7-O-glucoside (LUT-7G), a glycosylated form of luteolin, has been shown to exert anti-inflammatory, antioxidant, and antiproliferative properties by targeting the JAK/STAT3 pathway to inhibit STAT3 and down-regulating the expression of IL-1β, a target gene involved in inflammation and ROS production in ECs [58]. Other studies have also shown that luteolin inhibit the Gas6/Axl (growth-arrest-specific protein 6/tyrosine protein kinase receptor) pathway to resist angiogenesis [105] and may also activate the large conductance calcium-regulated potassium channel (mitoBKCa) to exert cardioprotective effects [106].

Baicalin is the most abundant flavonoid in the Chinese herb *Scutellaria baicalensis*, and baicalin is metabolized to baicalein by β-glucuronidase in the intestines, a metabolic process that is a key stage in the absorption of baicalin [107]. In terms of ameliorating vasodilatory dysfunction, baicalein (6.25–50 μmol/L) was able to activate the angiotensin-converting enzyme ACE2/Ang-(1-7)/Mas axis to reduce AngII levels, causing concentration-dependent vasodilation in Ang II-pretreated endothelium-intact aortic rings [59]. Notably, the vasodilatory function of baicalein may act directly on vascular smooth muscle through different Ca2+ channels as well as activated KATP channels rather than endothelium-dependently [108]. In terms of anti-oxidative stress, several studies have shown that baicalein attenuates NOX activity, down-regulates ROS levels, and up-regulates eNOS expression to increase NO production, which is often associated with activation of the PI3K/AKT/eNOS pathway [108,109,110]. Malondialdehyde (MDA), superoxide dismutase (SOD), and glutathione peroxidase (GSH-Px) are important indicators for assessing oxidation, and ROS in vivo can be catalytically inactivated and scavenged by SOD and GSH-Px [111]. Baicalein has been found to be effective in reducing MDA levels and increasing SOD and GSH-Px activities in rat lung tissues [112].The endothelin-1/endothelin A receptor (ET-1/ETAR) cascade has been suggested to be an important source of ROS, and the antioxidant activity of baicalein may be at least partly due to its inhibition of the ET-1/ET product, namely ROS formation [60]. Inflammatory coagulation dysfunction is based on inflammatory responses and platelet overactivation in endothelial dysfunction. Recent studies have shown that baicalein was able to inhibit both the Furin/TGFβ1/Smad3/thrombospondin-1 (TSP-1) pathway in ECs and the AKT/Ca2+/ROS pathway in platelets to ameliorate inflammatory coagulopathy [61]. It exerts anti-inflammatory effects by significantly inhibiting the expression of NF-κB, TNF-α, IL-6, and IL-1β [113,114]. In vitro experiments, baicalin was found to enhance AMPK-TFEB activity activating autophagy. Meanwhile, it was also able to increase the expression of the anti-apoptotic protein B-cell lymphoma 2 (Bcl2) and down-regulate the expression of the pro-apoptotic proteins BCL2-Associated X (Bax) and C-caspase3 to exert endothelial protective effects [115]. The protective effect of baicalin on the endothelium may also be reflected in the regulation of lnRNA NEAT1 as well as miRNA-205-5p [116]. In addition to medical experiments, baicalein has been applied to surface engineering of vascular scaffolds, and scaffolds with appropriate fixation densities of approximately 2.03 μg/cm^2^ successfully supported the growth of ECs and also modulated oxidative stress, inflammation, and hyperlipidemia in the pathological microenvironment, thereby inhibiting endothelial dysfunction [117].

Taken together, both luteolin and baicalin, which belong to the same family of flavonoids, exhibited pharmacological effects in terms of anti-inflammatory, antioxidant, and improvement of exogenously induced vasodilatory dysfunction.

### 3.2. Flavonols (Quercetin)

Among flavonoids, flavonols (together with flavanols) are by far the most abundant and widely distributed in nature, the typical flavonol being quercetin. Quercetin rapidly binds to glucuronic acid and/or sulphate during first-pass metabolism (entero-hepatic) so that the important metabolites of quercetin in human plasma are quercetin-3-glucuronide, quercetin-3-sulphate, and isorhamnetin-3-glucuronide. Quercitin’s bioavailability depends on the properties of the attached sugars and the food matrix composition (ethanol, fat, and emulsifiers), which may affect its solubility [118]. Interestingly, the bioavailability of quercetin glucoside from onions or processed onions is much higher than that of pure quercetin glycosides in capsules or tablets [119]. In in vitro assays involving HUVECs, quercetin exerted a wide range of anti-inflammatory properties. These include inhibition of the NFκB signaling pathway and down-regulation of the gene expression of inflammatory cytokines (e.g., IL-1β, IL-6, IL-8, and TNF-α) [62,63]. It is also able to inhibit the expression of E-selectin, MCP-1, VCAM-1, and ICAM-1 to prevent monocyte adhesion to endothelial cells [64]. It is important to note that in the large intestine, the catabolic metabolite of quercetin by the microbiota, 3-(3-hydroxyphenyl)propionic acid, exerts an anti-inflammatory effect through inhibition of TNFα-induced adhesion of monocytes to HAEC and inhibition of the up-regulation of E-selectin but is not involved in the regulation of ICAM-1/VCAM-1 [120]. However, in a novel in vitro multicellular model mimicking the intestinal-endothelial-monocyte/macrophage axis, quercetin intervention was able to reduce soluble vascular cell adhesion molecule-1 (sVCAM-1) levels, thereby improving the pro-inflammatory cellular environment [62]. In addition, it protects endothelial function from inflammation induced by local perturbation of blood flow, such as disturbed flow (DF), through inhibition of the NRP2-VEGFC (neuropilin 2–vascular endothelial growth factor C) complex [65]. Recent studies have shown that quercetin–lycopene combination (molar ratio 5:1) prevents oxidative stress in HUVEC cells by inhibiting the SIRT1-Nox4-ROS axis, reducing ROS production [66]. Its antioxidant effect is also manifested in enhancing the expression and activity of the antioxidant enzyme heme oxygenase-1 (HO-1), which has been confirmed in vivo/in vitro experiments [67]. In a human brain microvascular endothelial cells (HBMECs) injury model established by hypoxia/reoxygenation (H/R), quercetin (0.5–1μmol/L) can up-regulate the levels of Kelch-like ECH-associated protein 1 (Keap1) and nuclear factor erythroid-2 related factor 2 (Nrf2) to enhance antioxidant capacity [68]. The improvement of quercetin on diabetes-induced endothelial dysfunction is reflected in many aspects, including inhibition of endoplasmic reticulum stress-mediated oxidative stress [68,121,122], differential regulation of glucose uptake/metabolism in endothelial cells [123], and activation of autophagy [124]. The mechanisms of quercetin in alleviating endothelial senescence [125] and inhibiting EndoMT have been continuously excavated in recent years, the latter being mainly reflected in the inhibitory effects on mediators of EndoMT, such as TGF-β, Caveolin-1, NFκB, and ET-1 [126].

### 3.3. Flavanols (EGCG)

Flavanols are commonly found in fruits, green tea, red wine, chocolate, and other foods. The most common type of flavanols in nature is flavan-3-ols, which have a monomer form (catechin) and polymer form (proanthocyanidins) and other derivative compounds (such as theaflavins and thearubigins) [127]. Among green tea polyphenols, epigallocatechin gallate (EGCG) is considered to be the most abundant and active compound and has a wide range of therapeutic properties in cardiovascular and metabolic diseases, including anti-atherosclerosis, anti-diabetes, anti-inflammation, and antioxidants [128]. In an Ang II-induced hypertensive mouse model, EGCG (50 mg/kg/day) attenuates oxidative stress by down-regulating NOX expression and inhibits eNOS uncoupling to increase NO utilization, thereby exerting antihypertensive effects [69]. In vitro, EGCG can prevent homocysteine-induced oxidative stress and endothelial cell apoptosis by enhancing SIRT1/AMPK and Akt/eNOS signaling pathways [70]. Activated AMPK may also down-regulate the PI3K-AKT-mTOR pathway to induce autophagy, thereby achieving a protective effect on the endothelium [71]. Inhibition of inflammation by EGCG is mainly achieved by down-regulation of multiple components of the TNF-α-induced NF-κB signaling pathway, achieving inhibition of inflammatory gene transcription and protein expression [72]. In addition, the inhibitory effect of EGCG on EndMT has been confirmed, providing more possibilities for the prevention and treatment of cardiovascular diseases in the future [129]. The multi-type nanoparticle carriers developed for EGCG have been tested in simulated gastric and intestinal fluid environments to enhance their stability in acid–base and enzymatic hydrolysis environments, which is conducive to improving bioavailability in the human body [130,131].

### 3.4. Flavanones (Hesperidin and Naringin)

Flavanones, including hesperidin (hesperetin) and naringin (naringenin), are polyphenolic compounds highly and almost exclusively found in citrus. The protective effects of flavanones on the endothelium are mainly in terms of anti-inflammatory, antioxidant, and vasorelaxant properties, which are clinically important for vascular diseases such as hypertension and atherosclerosis [132]. Studies have shown that naringin (50, 100 μM) attenuated oxygenated low-density lipoprotein-induced inflammatory injury in HUVECs by down-regulating IL-1β, IL-6, and IL-18 levels. Meanwhile, VE-calmodulin, a specific dermatocyte marker for EndMT, was significantly attenuated, and serial evidence suggests that this may act by inhibiting the Hippo-YAP pathway [73]. Li et al. [74] confirmed that naringenin could activate AMPKα/Sirt1 signaling pathway, restore mitochondrial Ca2+ balance, reduce eNOS uncoupling, and increase NO bioavailability to improve damaged ECs by new transcriptomics technology. Its up-regulation of heat-shock protein 70 in endothelial cells has potential implications for the amelioration of endothelial damage in diabetes and related diseases [133]. Naringenin attenuates high-glucose-induced apoptosis in HUVECs, which may play a role in increasing HO-1 expression in HUVECs through up-regulation of PI3K/Akt or JNK pathways [75]. The vasodilatory effect of naringenin mainly involves the inhibition of multiple Ca2+ channels, whereas hesperidin stimulates a transient receptor potential vanilloid 1 (TRPV1)-mediated cascade reaction that activates the expression of both the CaMKII/p38 MAPK/MasR and the CaMKII/eNOS/NO signaling axes in HUVEC to promote NO production and vasorelaxant Mas receptor (MasR) expression [76,77]. Multiple nanodelivery systems increase the oral availability and targeting of compounds. For example, naringin/indocyanine green-loaded lipid nano-emulsions can effectively target VCAM-1 in the vascular endothelium to exert anti-inflammatory effects [134].

### 3.5. Isoflavones (Genistein and Daidzein)

The main plant source of dietary isoflavones is the soybean, with genistein (GE) and daidzein (DE) being the most abundant soy isoflavones, the latter being metabolized to estragole by intestinal flora [135]. Preliminary in vitro experiments have shown that Ge and De are able to be taken up by ECs and metabolized by phase II enzymes to methoxylated and glucuronide- and sulphate-conjugated compounds. However, the metabolite estragole was only taken up by ECs and not metabolized [136]. In an animal experiment, GE (40, 80 mg/kg) was used to treat two-kidney-one-clip (2k1c) hypertensive rats, which showed that GE inhibited ACE activity; attenuated Ang II, MDA, and SOD levels; and alleviated nephrogenic vasculopathy mediated by the RAS system [137]. In in vitro experiments, GE (1μM) can enhance the autophagy flux mediated by SIRT1/LKB1/AMPK pathway to prevent oxLDL-induced HUVECs senescence [78]. In another H_2_O_2_-mediated HUVECs senescence, GE (40, 80 μg/mL) exerted its anti-aging effect by inhibiting the TXNIP/NLRP3 (thioredoxin-interacting protein) axis [79]. In addition, GE was found to inhibit chronic vascular inflammatory responses in mice, which may be related to the inhibition of VEC inflammatory injury through the miR-21/NF-κB p65 pathway [138]. In previous studies, DE and estragole were found to have anti-oxidative stress effects, both involving inhibition of NF-κB proteins, with the latter further inhibiting endoplasmic reticulum stress, thereby attenuating atherosclerosis in ApoE-deficient mice [80,138,139].

### 3.6. Anthocyanins

Anthocyanins, as a natural pigment, are water-soluble and often abundant in fruits and vegetables such as blueberries, grapes, dragon fruit, purple sweet potatoes, and purple kale and are extremely important secondary metabolites in plants [140]. Anthocyanins are metabolized by the enterohepatic metabolism, where they are degraded to glucuronic acid and methylated and sulphated metabolites, i.e., phase II metabolites, and thereafter peak in plasma (<5 μM) in 1–3 h [141]. Unabsorbed anthocyanins are extensively metabolized to phenolic acids in the colon by the intestinal flora. Phenolic acids after methylation, such as vanillic acid (VA) or ferulic acid (FA), can be detected in plasma within 1 h after intake at peak concentration (2–15 μM) [142]. Various anthocyanins (usually in glycoside form) have been shown to increase cell viability by reducing ROS and NOX4 expression through a PI3K/Akt calcium-independent pathway in ECs. It has also been shown to promote catabolism of the protein kinase C zeta (PKCζ) pathway, increase eNOS and NO biomass levels, and reduce xanthine oxidase-1 (XO-1) and LDL levels to exert vasodilatory effects [81]. In a related study in aged rats, anthocyanins were found to modulate ROS formation and down-regulate SIRT1 to reduce eNOS uncoupling and prevent endothelial senescence by enhancing NO bioavailability [82]. It should be noted that the anthocyanin cornflower-3-glucoside (C3G), after 4 h of co-cultivation with epithelial cells, resulted in a loss of up to 96% due to an increase in the content of its metabolite protocatechuic acid (PCA) [82]. In addition, the C3G metabolite, i.e., PCA, with vanillic acid (VA) only caused a decrease in superoxide production and did not up-regulate eNOS levels, so it can be predicted that phenolic metabolites only increase the bioavailability of NO without increasing its production [143]. In terms of anti-inflammation, individual anthocyanins and their phenolic metabolites were able to attenuate monocyte adhesion to TNF-α-activated ECs at physiological concentrations (0.1–2 μM); however, this is not mediated by adhesion molecules such as VCAM-1 [144]. In addition, the mixture of various anthocyanins and metabolites (0.1 μM) can regulate the expression of EC permeability-related miRNAs, thereby reducing monocyte adhesion and transendothelial cell migration [145]. Studies have shown that anthocyanins can up-regulate antioxidant defenses such as HO-1 and SOD (in a concentration-dependent manner) through direct activation of the Nrf2 signaling pathway, which may prevent oxidative damage [146]. Transcriptomic and further analyses demonstrated that carrot-derived anthocyanins also inhibit palmitate-induced endothelial cell apoptosis via the p38 MAPK pathway [83]. In recent studies, genetic manipulation techniques have been applied to increase the anthocyanin content of plant-derived diets, mainly by up-regulating the expression of genes in metabolic pathways, which may also help to counteract the reduced bioavailability in vivo [147].

### 3.7. Chinese Herbal Medicine Flavonoids (Crocetin and Crocin)

For thousands of years, the wide application of Chinese herbal medicine, especially the compound decoction of multiple herbs, has played a great role in the prevention and treatment of cardiovascular diseases. With the development of medical information technology, flavonoids derived from Chinese herbal medicine have attracted more and more attention from researchers due to their significant clinical efficacy and relatively low drug toxicity.

Crocetin and its glycoside crocin are two important carotenoids isolated from the dried stigma of saffron. In the HUVEC model, crocetin (10, 20, and 40 μM) and crocetin (100, 200, and 400 μM) inhibited cell migration and angiogenesis and inhibited phosphorylation of vascular endothelial growth factor receptor 2 (VEGFR2) and its downstream pathway molecules (e.g., p-SRC, p-FAK, p-MEK, and p-ERK) [84]. Among them, molecular docking studies showed that crocetin showed stronger ability to bind to VEGFR2. In addition, a pharmacokinetic study showed that crocin is hydrolyzed into crocetin through the gastrointestinal tract and then absorbed and detected in plasma [148]. Occlusive zone-1 (ZO-1) is a scaffold protein in cerebrovascular endothelium, which is related to the function of blood–brain barrier. Loss of ZO-1 and disruption of the permeability of the blood–brain barrier (BBB) are easily observed in cerebral ischemia cases associated with oxidative stress and inflammatory response [149]. Hydroxy saffron yellow A (HYSA) is the main active ingredient of safflower. Studies have shown that HSYA protects ZO-1 from proteasome degradation by inhibiting the HIF-1α/NOX2 signaling cascade, thereby protecting cerebrovascular integrity and reducing brain damage [85]. In a study on the dilation activity of mesenteric artery rings (MAs) in rats, HYSA reversed the contraction of MAs induced by phenylephrine (PE, 1 μM) and U 46,619 (10 μM) in a dose-dependent manner. This function is related to the TRPV4-coupled Ca2+/PKA/eNOS signaling pathway, which achieves vasodilation by increasing Ca2+ influx [86]. In addition, flavonoids from other traditional Chinese medicines (TCM), such as ginkgolide B, have been found to stimulate SOD activity and increase SOD1 expression in diabetic aortas, thereby activating the Akt/eNOS signaling pathway and causing endothelium-dependent relaxation [150]. Since the application of herbs in TCM is often in the form of compounded soups and often in certain ratios, future pharmacological studies on flavonoid compounds in Chinese herbs may try to combine them in anticipation of obtaining higher efficacy and developing a wider range of drug targets.

## 4. Conclusions and Perspectives

The endothelium can secrete a variety of vasoactive factors, and it is not only the guardian of the cardiovascular environment but also the victim of cardiovascular disease. Traditional endothelial dysfunction involves inflammatory responses, oxidative stress, and its induction of platelet aggregation, autophagy, and apoptosis. Iron death [151], d-flow [152], and its associated EndMT [153] are considered as some of the key drivers of vascular inflammation and AS, which have been on the rise in recent years. As described in the text, natural flavonoid compounds or their metabolic components have been shown to target the endothelium through multiple pathways to exert regulatory effects and thereby ameliorate cardiovascular-related diseases. It is worth noting that TCM often applies herbs in disease treatment by combining multiple types of herbs in different dosages and preparing them according to specific methods, which may be informative for the exploration of enhanced pharmacological activities of flavonoids. In addition, improving the bioavailability of dietary flavonoids in vivo should still be a hotspot for future research, especially the further metabolism of intestinal flora and the further development of modern technological tools such as nanoparticles and vascular scaffold surface engineering.

## Figures and Tables

**Figure 1 pharmaceuticals-16-01201-f001:**
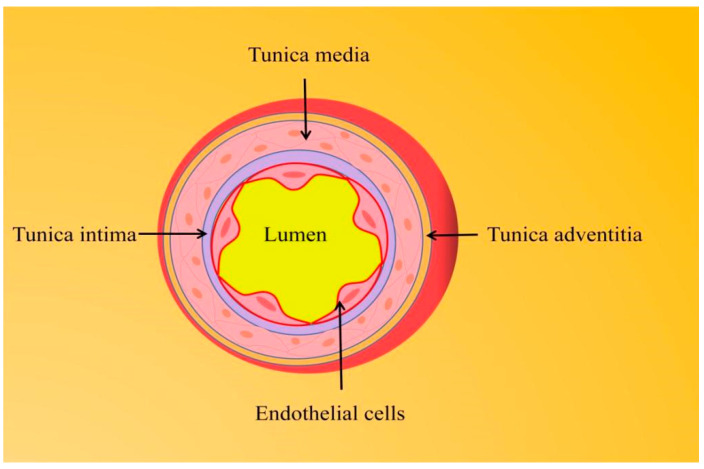
The arterial vessel wall is composed of three layers from outer to inner, namely the adventitia, media, and intima.

**Figure 2 pharmaceuticals-16-01201-f002:**
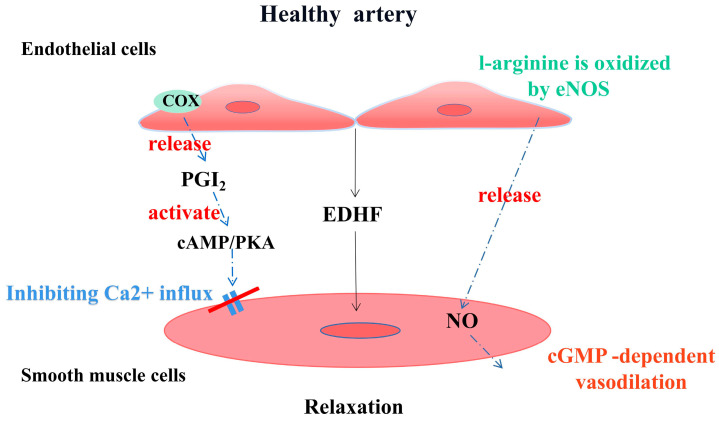
Endothelial-derived vasoactive factors regulate vascular tone in healthy arterial vessels.

**Figure 3 pharmaceuticals-16-01201-f003:**
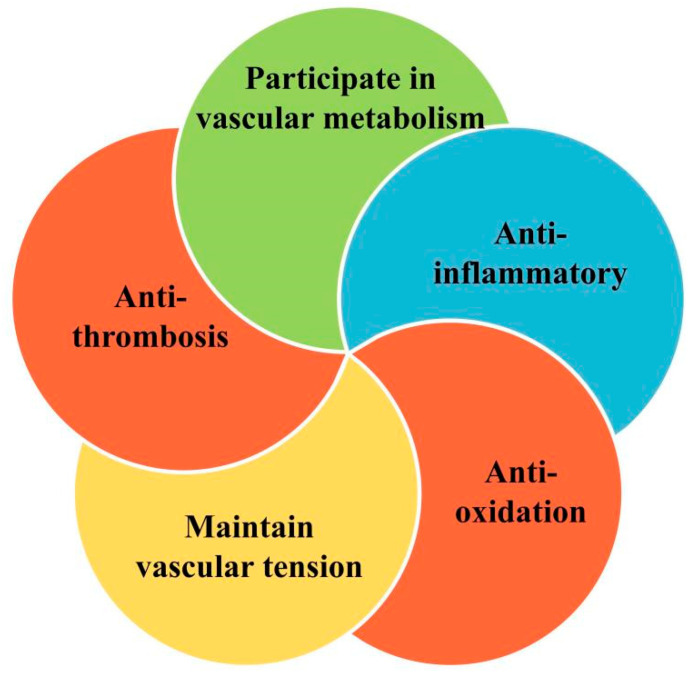
Regulation of vascular homeostasis by healthy endothelium.

**Figure 4 pharmaceuticals-16-01201-f004:**
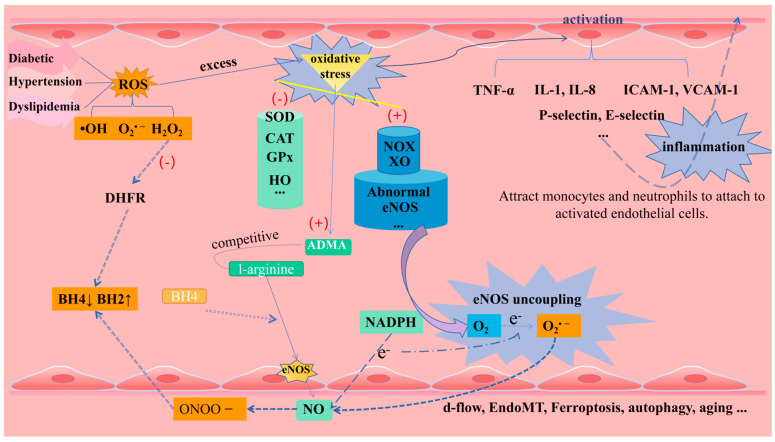
Endothelial dysfunction (including oxidative stress, eNOS uncoupling, inflammation, and other factors).

**Figure 5 pharmaceuticals-16-01201-f005:**
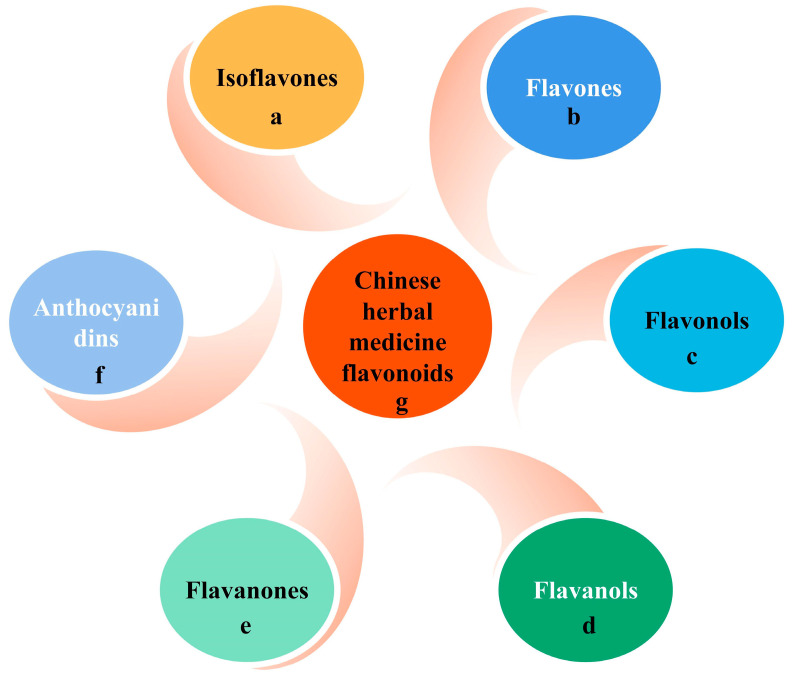
Flavonoids in dietary and herbal medicines.

**Figure 6 pharmaceuticals-16-01201-f006:**
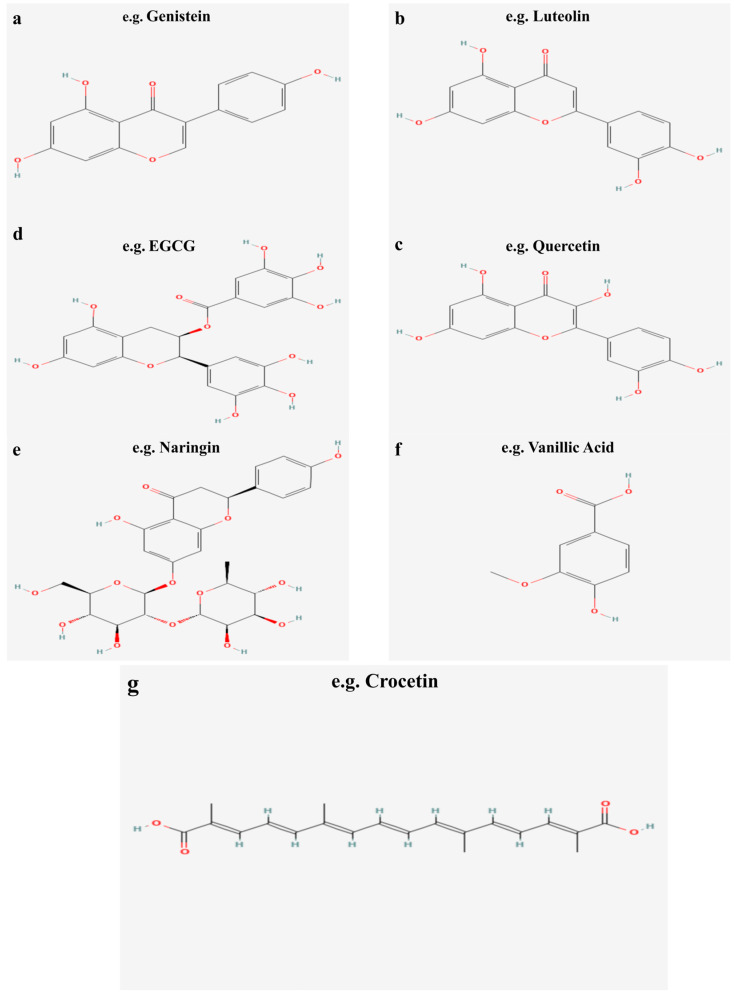
Molecular structure of typical flavonoids in dietary and herbal medicines.

**Figure 7 pharmaceuticals-16-01201-f007:**
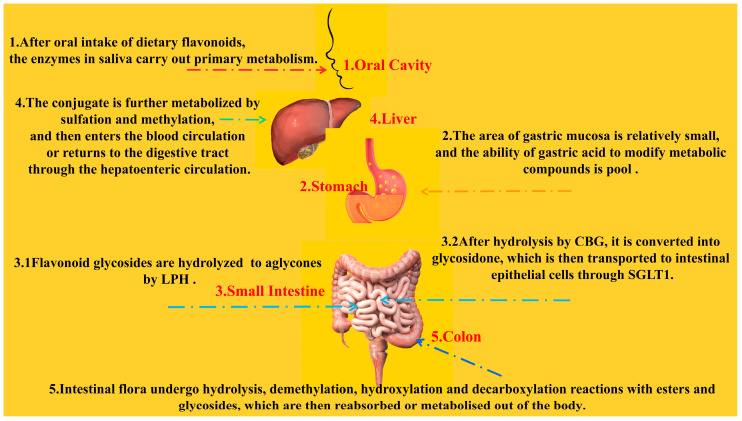
Metabolic pathways of flavonoids in the human body.

**Table 1 pharmaceuticals-16-01201-t001:** Endothelial-protective mechanisms of flavonoids in vivo and in vitro. **↑** for up-regulation, and **↓** for suppression.

Model	Components	Dose	Function	Signal Passage	Ref.
Hypoxia-induced pulmonary hypertension in rats	Luteolin	10–100 μmol/L, 28 days	Aortic ring relaxation; Mean pulmonary arterial hypertension ↓;	HIF-2α-Arg-NO axis ↓ and PI3K-AKT-eNOS-NO ↑	[54]
H_2_O_2_-induced injury of HUVECs	Luteolin	2.5–20 μM, pretreatment 2 h	Anti-oxidative stress; improves mitochondrial function	AMPK/PKC ↑; P38 MAPK/NF-κB ↓	[55,56]
TNF-α-induced adhesion of human EA.hy 926 ECs	Luteolin	0.5–20 μM, pretreatment 1 h	(MCP-1, ICAM-1, VCAM-1) ↓	IKBα/NF-κB ↓	[57]
TNF-α-induced C57BL/6 mice	Luteolin	Modified diet containing 93.93% luteolin	Anti-inflammatory	IKBα/NF-κB ↓	[57]
HUVECs	Luteolin- **7-O-Glucoside**	20 μL, treatment for 48 h	Anti-oxidative stress; Anti-inflammatory; Anti-proliferation	JAK/STAT3↓; Nox4/ROS-NF-κB↓; MAPK ↓	[58]
AngⅡ-induced injury of HUVECs	Baicalin	6.25–50 μM	Anti-oxidative stress; Anti-apoptosis	Activation of the ACE2/Ang- (1-7)/Mas axis; PI3K/AKT/eNOS ↑	[59]
Norbascine-induced pulmonary hypertension in rats	Baicalein	10 mg/kg/day, 28 days	Anti-oxidative stress; Mean pulmonary arterial hypertension ↓	Akt/ERK1/2/GSK3β/ β-catenin ↓; ET-1/ETAR ↓; ROS ↓	[60]
LPS-induced rats	Baicalin	50, 100 mg/kg/day, 3 days	Inhibited platelet hyperactivation; Anti-inflammatory; TSP1 ↓	Furin/TGFβ1/Smad3/TSP-1 ↓	[61]
TNF-α-induced injury of HUVECs	Baicalin	/	Anti-platelet adhesion; TSP1, ICAM-1 ↓	AKT/Ca2+/ROS ↓	[61]
TNF-α-induced injury of HUVECs	Quercetin	10 μM; 30 μg/mL	Anti-inflammatory; anti-apoptosis E-selectin, VCAM-1, ICAM-1, IL-6, IL-8 ↓	Activator protein 1 (AP-1) ↓ NF-κB ↓	[62,63,64]
DF-induced inflammation of HUVECs	Quercetin	5 μM	Anti-inflammatory	NRP2 -VEGFC complex ↓	[65]
H_2_O_2_-induced injury of HUVECs	Quercetin–lycopene combination (molar ratio 5:1)	8 μM, 12 h	Anti-oxidative stress; Anti-inflammatory	SIRT1-Nox4-ROS ↓	[66]
High-fat diet (HFD)-fed ApoE^−/−^ mice	Quercetin	4 mg/day, 8 weeks	Anti-oxidative stress	NOX ↓; HO-1↑	[67]
H/R-induced injury of HBMECs	Quercetin	0.1–1 μmol/L, 8 h	Anti-oxidative stress; Enhance cell viability; Anti-apoptosis; ICAM-1, VCAM-1 ↓	Keap1/Nrf2 ↑	[68]
AngⅡ-infused hypertensive mice	EGCG	50 mg/kg/day	Anti-oxidative stress; Systolic blood pressure ↓	NOX ↓; BH4-eNOS-NO ↑	[69]
Homocysteine-induced injury of HUVECs	EGCG	Pretreatment 2 h	Anti-oxidative stress; Anti-apoptosis	SIRT1/AMPK ↑; Akt/eNOS ↑	[70]
H_2_O_2_-induced injury of HUVECs	EGCG	1–10 μmol/L, pretreatment 24 h	Anti-oxidative stress; Induced autophagy	PI3K-AKT-mTOR ↓	[71]
TNF-α-induced injury of human coronary artery endothelial cells (HCAECs)	EGCG	/	Anti-inflammatory	NF-κB ↓	[72]
ox-LDL-induced injury of HUVECs	Naringin	50, 100 μM, pretreatment 2 h	Anti-inflammatory; Anti-apoptosis; Anti-EndMT	Hippo-YAP ↓	[73]
Homocysteine-induced injury of HUVECs	Naringenin	200 μM, treatment for 24 h	Anti-oxidative stress; Reduced eNOS uncoupling	AMPKα/Sirt1 ↑	[74]
High glucose (HG)- or free fatty acids (FFA)-induced apoptosis in HUVECs	Naringenin	0–100 μM	Anti-apoptosis	PI3K/Akt and JNK1 ↑; Nrf2 ↑; HO-1 ↑	[75]
HUVECs	Hesperidin	1 μM, 2 h	Promoted NO production and expression of MasR	TRPV1-CaMKII/p38 MAPK/MasR ↑; TRPV1-CaMKII/eNOS/NO ↑	[76,77]
Ox-LDL-induced senescence of HUVECs	Genistein	1 μM, pretreatment 30 min	Induced autophagy; Anti-aging	SIRT1/LKB1/AMPK ↑	[78]
H_2_O_2_-induced senescence of HUVECs	Genistein	40, 80 μg/mL, 24 h	Anti-apoptosis; Anti-aging	TXNIP/NLRP3 ↓	[79]
LPS-induced chronic vascular inflammatory response in mice	Genistein	10 mg/kg/day, 20 weeks	Anti-inflammatory	miR-21/NF-κB p65 ↓	[80]
Vascular endothelial cells (VECs)	Genistein	10 μM, pretreatment 2 h	Anti-inflammatory	miR-21/NF-κB p65 ↓	[80]
High glucose (HG)-induced injury of HUVECs	Blueberry anthocyanins	5 μg/mL, pretreatment 24 h	Anti-oxidative stress; Induced vasodilation	PI3K/Akt ↑; PKCζ ↓	[81]
Aged SD rats	Mulberry extract	300 mg/kg	Anti-oxidative stress; Anti-aging; Reduced eNOS uncoupling	SIRT1 ↑;	[82]
PA-treated SV 40 transfected aortic rat endothelial cells (SVAREC)	Anthocyanin from red radish	50, 100, 200, 400 μ g/mL, 24 h	Anti-apoptosis	p38 MAPK ↓	[83]
HUVECs	Crocetin	10, 20, 40 μM	Inhibited cell migration and angiogenesis	VEGFR2/SRC/FAK ↓	[84]
HUVECs	Crocin	100, 200, 400 μM	Inhibited cell migration and angiogenesis	VEGFR2/MEK/ERK ↓	[84]
LPS-stimulated brain microvascular endothelial cells	Hydroxysafflor Yellow A	/	Prevented ZO-1 degradation and protected the blood–brain barrier	HIF-1α/NOX2 ↓	[85]
U-46,619- and PE-inhibited rat MAs	Hydroxysafflor Yellow A	10^−7^, 10^−6^, 10^−5^, 10^−4^ M	Increased Ca2+ influx and expanded blood vessels	TRPV4-coupled Ca2+/PKA/eNOS ↑	[86]

## Data Availability

Data sharing not applicable.

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
