# Peer review of "Research Progress of Flavonoids Regulating Endothelial Function"

_pharmaceuticals, 2023, doi:10.3390/ph16091201_

Round 1
Reviewer 1 Report
The present manuscript reviewed the functions and potential mechanisms of flavonoids in promoting endothelial dysfunction and cardiovascular diseases, and also introduced their bioavailability, to some extent, which is of significance. However, there are also several points in the manuscript that have to be carefully addressed as follows:
1. The introduction needed to be further refined. First, it is necessary to delete the content of line 70-92 (i.e. 1.1. Effects of dietary polyphenols on cardiovascular health: an epidemiological study); secondly, the content of subtitles “2” and“3”should be deleted properly and focus on the title, not state a lot “Endothelium and CVD” or “Flavonoids”.
2. Some classical flavonoids such as luteolin, baicalin, quercetin hesperidin, and naringin, etc. were reviewed, and it was not comprehensive enough.
3. In recent years, metabolomics and proteomics techniques have been widely used to explore the effects of flavonoids or traditional Chinese herbs abundant in flavonoids (sophora japonica, ginkgo biloba, safflower, etc.) on cardiovascular diseases and inflammation, which should be reflected in the paper.
4. It is suggested to reject the manuscript due to numerous revisions.
Minor editing of English is required.
Author Response
Dear reviewer,
I have revised the article according to your suggestions and attached a revision note, please find the details in the attached document.

Reviewer 2 Report
Reviewers' comments:
Manuscript ID: pharmaceuticals-2535719
Title: Research progress of flavonoids regulating endothelial function.
Manuscript Type: Review
Reviewers' comments:
The manuscript describes the Research progress of flavonoids regulating endothelial function. The manuscript needs a detailed editing. Some markings are made to just illustrate the extent of editing needed. A thorough revision addressing all the concerns is needed and if the authors are prepared to do that it can be considered for a review of the revised manuscript.
The authors need to consider the following comments
- In the Abstract, the authors need to improve.
- Add more suitable keywords.
- Figure 2. Endothelial-derived vasoactive factors regulate vascular tone in healthy and pathological arterial vessels – not clear make clear.
- 2.2.2. Inflammation - section should be detailed.
- 3.1. Chemical properties of flavonoids - should be detailed.
- 3.2. Bioavailability of flavonoids - section should be detailed.
- Figures 5 and 6 – not clear make clear.
- In the Conclusions and perspectives, the authors need to improve with more specific short results and conclusions.
- References: make all references in same format for volume number, page number and journal name, because it is difficult to searching and reading.
- Furthermore, they should add the graphical abstract, it is use full to readers.
So that I recommended this manuscript to major revision and for future process.
Minor editing of English language required
Author Response

(The authors gave the same response as above.)

Round 2
Reviewer 2 Report
The manuscript can have published. The authors have answered the questions.
Minor editing of English language required.